# Potential Barriers to the Implementation of Computer-Based Simulation in Pharmacy Education: A Systematic Review

**DOI:** 10.3390/pharmacy11030086

**Published:** 2023-05-17

**Authors:** Ahmed M. Gharib, Gregory M. Peterson, Ivan K. Bindoff, Mohammed S. Salahudeen

**Affiliations:** School of Pharmacy and Pharmacology, University of Tasmania, Hobart, TAS 7005, Australia

**Keywords:** computer simulation, pharmacy education, distance education, patient simulation, teaching, online learning, virtual reality, serious gaming, standardized patient

## Abstract

Computer-based simulation (CBS) is an interactive pedagogical training method that has seen increased interest, especially in recent years. There is some evidence that CBS in pharmacy education is not as widely adopted compared to other healthcare disciplines. Pharmacy education literature to date has not specifically discussed the potential barriers which may cause this uptake challenge. In this systematic narrative review, we attempted to explore and discuss potential barriers that may impact the integration of CBS in pharmacy practice education and provide our suggestions to overcome them. We searched five major databases and used the AACODS checklist for grey literature assessment. We identified 42 studies and four grey literature reports, published between 1 January 2000 and 31 August 2022, which met the inclusion criteria. Then, the specific approach of Braun and Clarke for thematic analysis was followed. The majority of the included articles were from Europe, North America, and Australasia. Although none of the included articles had a specific focus on barriers to implementation, thematic analysis was used to extract and discuss several potential barriers, such as resistance to change, cost, time, usability of software, meeting accreditation standards, motivating and engaging students, faculty experience, and curriculum constraints. Ad- dressing academic, process, and cultural barriers can be considered the first step in providing guidance for future implementation research for CBS in pharmacy education. The analysis suggests that to effectively overcome any possible barriers to implementing CBS, different stakeholders must engage in careful planning, collaboration, and investment in resources and training. The review indicates that additional research is required to offer evidence-based approach and strategies to prevent overwhelming or disengaging users from either learning or teaching process. It also guides further research into exploring potential barriers in different institutional cultures and regions.

## 1. Introduction

Pharmacy education incorporates various simulation methodologies (such as computer-based simulations (CBS), high-fidelity manikins, medium-fidelity manikins, standardized patients, roleplaying, and Objective Structured Clinical Evaluations (OSCE)) [1,2]. Notably, CBS stands out for being more scalable, easily accessible, and cost-efficient and offers a standardized experience for pharmacy students [3,4,5,6,7,8,9]. In addition, CBS provides remote access [10], self-directed and self-paced learning [10,11,12], instant feedback [13], and easily updatable content [14,15]. Several studies have also illustrated that CBS can establish a connection between theoretical knowledge and practical application, supplement- ing scarce placement options by enabling students to practice their professional roles repeatedly and on-demand [6,16,17,18,19,20,21,22]. In addition, CBS has proved to be as capable in developing students skills [23] (both technical [17,24,25,26] and non-technical [17,24,27] as well as dispensing skills competency [28]. Moreover, CBS fosters the growth of students’ clinical decision-making abilities by employing evidence-based practice in a dynamic learn- ing setting, focusing on knowledge application rather than conventional memorization- based education [29,30]. Consequently, CBS can function as both an educational resource and an evaluation instrument [13]. Along with these benefits, the current generation of learners, considered by many as “digital natives”, typically expects the extensive use of technology in their education [31,32,33,34,35]. Moreover, the unexpected COVID-19 pandemic has necessitated a change in pharmacy education [36,37,38], leading many education providers to implement, study, and improve CBS as a training method [10,39,40,41].

Despite the potential for integrating CBS into pharmacy practice education, its adoption in pharmacy education seems less widespread compared to other healthcare disciplines. Conversely, the majority of CBS research has been conducted in the fields of medicine or nursing [35,42]. There is also limited research indicating methods for the effective implementation of CBS in pharmacy education [43,44,45]. Most of the available literature is focused on high- and medium-fidelity mannequin simulations and standardized patients [46,47,48,49]. For instance, Kane-Gill and colleagues identified only two articles related to CBS among 13 studies that examined the transfer of knowledge from simulation to pharmacy practice [44]. Another study by Jabbur-Lopes and colleagues found a mere seven out of 263 potential studies concerning the use of virtual patients in pharmacy education [16]. Regrettably, a significant gap exists in the literature when it comes to recognizing and exploring the perspectives of key stakeholders (educational organizations, educators, and students) on potential barriers affecting the implementation process and adoption of CBS in pharmacy practice education. Distinct stakeholders may have varying needs, which could directly or indirectly influence the success or failure of the implementation process [50]. It is believed that understanding this would enable a better-informed implementation strategy [51,52]. Consequently, this review aimed to identify and comprehend potential barriers that might affect the implementation of CBS in pharmacy practice education and guide future research in this area. In addition, it supplements our recently conducted review where we sought to examine different features of available simulators in pharmacy practice education [53].

## 2. Methods

In this review, a systematic narrative review methodology was adopted as it facilitated the inclusion of quantitative, qualitative, and mixed-methods literature, as well as the inclusion of grey literature. This methodology seemed the most suitable to extract and summarize potentially identified barriers. We adhered to the guidelines for systematic reviews and meta-analyses as suggested by the Preferred Reporting Items for Systematic Reviews and Meta-Analyses (PRISMA) to conduct this narrative systematic review [53,54,55,56,57] (Appendix A).

### 2.1. Data Sources and Search Strategy

In order to achieve the research objectives, a literature search was carried out utilising five key databases: Ovid Medline, CINAHL, ERIC, Education Source, and Ovid EMBASE. The chosen timeframe for the articles spanned from 1 January 2000 to 31 August 2022. Moreover, a grey literature search was conducted to minimize publication bias, enhance the review’s thoroughness, and provide a well-rounded representation of the existing evidence. The grey literature search included relevant conference abstracts, proceedings, and reports. This search also included Google Scholar, Trove and Bielefeld Academic Search Engine (BASE). The search was supplemented by reviewing the reference lists of the included studies and their citations to discover any additional resources.

The search terms used in this study were ‘pharmacy education’; ‘teaching method’; ‘computer-based simulation’; ‘online simulation’; ‘virtual simulation’; and ‘virtual patient’ (the search strategy is shown in Appendix A). It should be noted that this search strategy was also used in our previously published paper which identified the range and characteristics of CBS available in pharmacy education [53].

### 2.2. Screening and Selection Criteria

The entire list of citations obtained through the search strategy was imported into Covidence [58], and studies that were duplicated were omitted from the review. The citations were then subjected to a two-phase screening procedure (title/abstract and full text). Two authors (AG and MS) independently reviewed each potentially relevant article to determine its eligibility for inclusion. Any disagreements were settled by a third author (IB/GP), with decisions being reached through consensus.

Overall inclusion criteria were (1) literature that focused on pharmacy practice education “tasks encountered in pharmacy practice and/or patient encounter experience”, (2) published between 1 January 2000 and 31 August 2022, and (3) focused on the implementation of CBS and mentioning barriers/facilitators. In addition to this, the included grey literature had to be (1) publicly available and (2) meet the Authority, Accuracy, Coverage, Objectivity, Date, Significance (AACODS) checklist [59]. Exclusion criteria were (1) non- English publications and (2) literature that addressed different forms of simulations (e.g., mannequins, roleplaying). In addition, low-credibility grey literature, such as anonymous publications, blogs, emails, letters, news, and wiki articles, were excluded.

### 2.3. Data Extraction and Synthesis

Two authors (AG and MS) independently carried out a comprehensive review of the included papers. All collated data were organized and presented in a table format. The collected information from the standard literature included general identifying information about the study (authors and year); country; participants; the simulation tool in use; and factors identified as barriers (can be seen in Appendix A). Extracted data from the included grey literature were based on the AACODS checklist and consisted of the following information: general information about the study (author, year, and title); authority; accuracy; coverage; objectivity; date; significance; and overall authors’ comments on CBS implementing in pharmacy education (can be seen in Appendix A). The data extraction was carried out by one researcher and cross-checked by another. Upon the completion of this process, the researchers had several discussions to collate the findings and interpret the outcomes. However, a meta-analysis could not be performed due to the significant heterogeneity of the information gathered.

The authors defined the barriers as ‘any factors that may obstruct the implementation process either directly or indirectly.’ The barriers to implementing new technologies in general education literature can often be grouped into three key areas: cultural, process, and academic [60]. Benedict et al. defined it as follows [12]: cultural barriers relate to attitudes toward technology and curricular change, process barriers address workflow, and academic barriers are related to technology meeting course aims and enhancing educator ability. Addressing these barriers can facilitate effective technology integration in the classroom and improve learning outcomes. We also attempted to analyze through the lens of the various stakeholders (educational organizations, educators, and students).

Given the diversity of the papers in the review, with their variety of methodologies, themes were grouped and addressed in the narrative synthesis of the literature [61]. The reviewed studies presented a range of different simulators (24 in total), each with its unique features, applications, and evaluation methods. These simulators were documented at various stages of their evolution, each providing different features and design approaches. Owing to these differences and considering that the barriers for one simulator may not be the same as for another, this exploratory review focused on only the overarching barriers. Thematic analysis was utilized to derive the primary themes related to these pervasive barriers that affect the integration of CBS [62]. The authors convened to discuss the principal themes that emerged.

To adopt a suitable structured thematic analysis for this review, the specific approach of Braun and Clarke (2006) was followed [62]. This approach is normally used with collected qualitative data rather than the texts of formal publications, but the clear guidelines allowed for a deliberate and rigorous thematic analysis to be undertaken, which suited the aims of this study [62]. An inductive approach was adopted to allow the data to determine the study themes, which involved following six main phases as a guideline. The first phase was “familiarization”, which involved checking the data and noting any initial observations for analysis. The second phase was “coding”, which involved capturing data concepts through coding. The third phase was “generating themes” by searching for themes and coding the data relevant to each theme. The fourth phase was “reviewing themes”, where relevant themes were included, and those out of scope were discarded. The fifth phase was “defining and naming the themes” to efficiently reflect the included data. Finally, the last phase involved “writing up” the analytic narrative and “contextualizing” it in relation to the existing literature. This approach helped to identify and organize key themes in the data and provided valuable insights into the research topic.

In phase I, familiarization was performed by carefully reading through the text, taking initial notes, and developing a general understanding of the several factors. Followed by phase II, shorthand labels or codes were developed, with each code describing a factor that was felt to be relevant or potentially interesting to consider as an influencing factor for the selection or implementation processes. This was done with every included study. Then, in phase III, all collected codes formed main groups to establish a condensed over- view of themes. In phase IV, usefulness of the themes and accuracy of presenting the data were checked by two reviewers (MS, IB) to ensure that we had captured and presented the identified themes accurately. This was followed by a discussion amongst the research team. In phase V, the reviewers agreed upon the final naming of each theme so that each theme reflected their included data efficiently and concisely. Finally, in phase VI, the authors wrote a narrative thematic analysis to address the research question and aims, as discussed in the findings section.

## 3. Results

The initial search yielded 1500 studies in total. Out of these, 43 were deemed suitable for a full-text review. Post full-text analysis, an additional 5 studies were removed as they did not align with the pre-established inclusion criteria (they were either not in English or addressed different forms of simulation). However, through citation analysis, 8 more potential studies were discovered. In the end, 46 studies were selected for inclusion in the qualitative synthesis (Figure 1).

### 3.1. General Characteristics of the Included Papers

This review included 46 pieces of literature; 41 studies from standard literature (89.1%: T1), one commentary [4], and four grey literature reports [2,18,63,64]. The included studies had a broad global representation; 46.3% (*n* = 19) of the standard literature studies were undertaken in the USA [5,6,7,12,39,65,66,67,68,69,70,71,72,73,74,75,76,77,78], followed by Australia (19.5%, *n* = 8) [3,4,28,79,80,81,82,83], then the UK (*n* = 4) [8,42,84,85], New Zealand (*n* = 2) [86,87], Canada (*n* = 2) [88,89], Brazil (*n* = 1), Sweden (*n* = 1) [9], Portugal (*n* = 1) [90], the Netherlands (*n* = 1) [91], and three studies that included participants from different countries (one in Australia and Malaysia [92], and the other for multiple European country graduates [35,93]).

A consistent number of papers related to the application and introduction of CBS in pharmacy education were published until 2020, when a noticeable surge was noticed. The majority of the studies employed online surveys post-course or post-assignment as their primary data collection technique, though a few incorporated both pre- and post-surveys [6,12,69,82,94]. Alternative data collection methods comprised focus groups [80,86,89,92,95], subjective-objective assessment plans (SOAPs) [66,96], and observations [9,73,86,90,95]. The majority of the participants in the included studies were undergraduate pharmacy students, with a few postgraduate students or practicing pharmacists.

### 3.2. Summary of Identified Barriers

The review identified eight key overarching barriers, which were categorized based on three primary themes (cultural, academic, and process-related barriers) and based on the stakeholder group that reported them (educational organizations, educators, and stu- dents). The identified overarching themes were resistance to change, cost, time, usability of software, meeting accreditation standards, motivating and engaging students, faculty experience, and curriculum constraints (Table 1).

#### 3.2.1. Cultural Barriers

This barrier group was the most reported barrier theme, and it directly deals with the adoption challenges of CBS, specifically concerning attitudes toward technology and making changes to curriculum. The overarching theme in the cultural group was resistance to change, where it was most reported under the educators’ group, followed by the educational organized, and then the students’ group.

Some educators reported resistance to adopting the CBS training method as it would require extensive technical skills and knowledge outside their traditional scope of practice [5,6,9,12,35]. In addition, some educators still viewed CBS as an experimental training method and felt that CBS is not adequately reliable or does not match the quality of traditional training [5,35]. Moreover, some studies reported that some educators were skeptical about how practical CBS would be for students’ training when tested in real-life situations [5,6,9,12,35].

Students’ prejudgment, attitudes, and resistance to change were reported in some studies (15 of those seen in Table 1). In some of those studies, it was reported that students were found to be skeptical about using CBS based on previous experience, misconceptions, or fears that such a training method would hinder their training quality. This pre- judgment among students may have affected their confidence in CBS [6,7,8,12,28,35,66,71,72,77,78,79,82,87,97], especially if they experienced system errors and/or technical difficulties [6,8,28,77,79]. Moreover, it was reported that some students might prefer the traditional method rather than adopting a new learning method [6,7,12,35,71,72,82].

#### 3.2.2. Process Barriers

This barrier group includes issues around usability, cost, and any other operational challenges. This was the second most reported barrier group. We found that students were concerned about the potential time taken to complete a scenario, while educator groups were also concerned about the time required to design a scenario, as well as general concerns around software errors and usability. In contrast, the educational organization group focused primarily on the overall costs associated with the use of educational software.

Cost was a common theme, with some institutions reporting a lack of resources [4,6,12,69,80,94]. Cost can also include ongoing costs associated with the development, implementation, and maintenance of CBS and their operations. For example, some studies identified high purchase fees or development costs as barriers to implementing CBS [5,6,35,39,69,94]. It may also include the cost needed for technical support associated with developing and maintaining the software itself (design, programming, and graphics) [4,5,39] or the need for additional third-party help, either in the form of hiring an experienced programmer or a specialized IT firm in educational simulations [9,39,74]. Establish- ing technical support infrastructure was seen by some as a burden with potential additional costs, involving staffing and specialized training (e.g., trained IT team for setting up the server and maintaining it) to ensure smooth operation and minimize error frequency, server overload, or system lag [2,5,28,39,94].

Another reported potential barrier was the software usability itself, with some papers reporting software-related challenges, such as the interface, ease of access, and general usability of the software for both students and educators [8,28,35,77,82]. This included difficulties in system navigation, data entry, complicated interface, system errors, glitches, lagging, and server overload [6,7,8,28,39,70,73,74,75,76,77,79,82,85], which required either CBS de- sign revision or software update.

Time was also reported as a barrier, particularly with regard to developing, editing, or updating scenarios. For example, when educators have to develop a new scenario to fit their course needs, which is a technically demanding and potentially time-consuming process; depending on the tools available and the complexity of the scenario, this can take anywhere from 8 to 100 h per case [5,6,7,12,28,39,65,66,69,72,74,78]. In addition, when educators decide to use a shared scenario design, they might spend significant time navigating the software without finding a match to their needs or may find a scenario that would require some updating or alteration, which is a potentially time-consuming and resource-intensive procedure [5]. Some of the shared scenarios are likely to be foreign to the local healthcare system in which a given educator is likely to operate, which would then require an understanding of the differences in laws, regulations, standards, and markets between different regions [7,70].

Similarly, time was reported as a student barrier in the context of the time needed by students to complete exercise content [72]. Despite some students feeling that case scenarios were time-consuming, studies have reported improvements in knowledge, skills, and enjoyment, indicating no negative impact on learning outcomes [7,28,39,70,73,75,76,82,85]. In addition, for some simulators, grading was not part of their design, requiring tutors and educators to engage in manual grading activities. Therefore, grading and case scenario development, editing, or updating can be identified as potential barriers affecting the educators’ workflow [5,6,7,66,69,72,74,77,80].

#### 3.2.3. Academic Barriers

This barrier group pertains to the educational and technical challenges that may affect the implementation of CBS, such as meeting accreditation requirements, faculty expertise and/or curriculum constraints, and student engagement.

Accreditation standards play a critical role in ensuring the quality of pharmacy education programs. Despite the recognition of CBS as a valuable tool in pharmacy education [2,64], meeting accreditation standards can sometimes present a barrier to its implementation. Whereas accreditation standards may recognize the importance of CBS in pharmacy education, the concept is often poorly defined, and implementation guidelines are not provided [64]. This may lead to confusion and uncertainty among educators who are trying to integrate CBS into their curricula.

Failing to meet students’ needs and expectations can also be a barrier [8,12,39,68,72,82,84,88,89,97]. Some studies have found that while simulated scenarios designed by CBS were close to reality, some students found the avatar’s appearance and interactions to be unrealistic [28,76,81,82,97].

Lack of technical expertise also presents as an academic barrier, where pharmacy educators may not have the skills required to develop the computer-based scenarios or sup- port the implementation of the CBS into the curriculum. This could lead to a lack of confidence in using the technology, difficulty in creating effective simulations, and difficulty in supporting students using the software [6,8,12].

In addition, pharmacy programs often have strict curricular requirements, especially regarding assessment, which may leave little room to incorporate new teaching methods such as CBS. It can be difficult to assess student performance in CBS, particularly if the simulations are not directly aligned with existing assessment methods. This could lead to difficulties in grading and evaluating student progress [7,72,74,80]. The simulator should ideally fit into any course based on how it can help in achieving the expected course outcomes [5,18,64,94]. Unfortunately, some simulators do not align with course needs and aims as expected, resulting in failure to meet the expected outcome of transferring knowledge [71,72].

## 4. Discussion

Implementing modern technologies or training methods into any curriculum can be challenging and may require significant changes. There may be resistance to changing established teaching practices, and educators may be unsure of how to integrate simulations into their existing course content. This review identified potential barriers to the adoption of CBS, many of which are similar to those reported in general education literature, such as cost, resource availability, willingness to change, resistance to change, staff training, student engagement, and workload concerns [98,99,100,101,102,103]. However, appropriate change management strategies may be able to mitigate many of these barriers. Furthermore, some of the identified barriers may not even be barriers if managed or approached differently; these include cost, educators’ resistance to change, and technical issues.

One of the common themes was concerns about cost. However, some studies considered the CBS that they were evaluating to be cost-efficient, especially in terms of running costs, when compared to traditional training methods [3,6,7,8,79]. In addition, some researchers have highlighted that creating new scenarios with CBS may be more cost- and time-effective compared to other training methods, such as roleplay or high-/medium fidelity manikins [7,76]. Some studies have looked at establishing a home-grown tool [2,5,28,39,94]. In these cases, the CBS would be a costly solution, with the cost of development and ongoing maintenance being borne entirely by the organization. However, other studies have explored the use of commercial products, where the costs of development and maintenance are external and distributed.

Resistance to change was another common theme. CBS can be a valuable addition to pharmacy educators’ teaching toolkits, offering various learning experiences, and aiding theory-to-practice transfer. However, implementing CBS may require significant time and effort to develop tailored scenarios for specific courses [5,6,7,12,28,39,66,69,72,74]. Developing an engaging and challenging content can be difficult and even frustrating [77,82,94]. This presents a significant barrier to CBS adoption.

However, some have argued that this is a one-time investment, as in the following years, once enough scenarios are produced, the modification would be minimal [72]. In addition, when the burden of scenario creation can be distributed across a global network of educators, the overall process may even be easier to administer than traditional alter- natives [72,82,104,105]. Many modern simulators offer a collaborative sharing feature, allowing their users to share scenarios [106,107]. Once the initial hurdles have been overcome, preparation time is expected to decrease dramatically [12,39].

In the literature, usability and technical issues were commonly identified, including a lack of training and difficulties in using the tools, although few studies provided specific examples on such experiences [28,70,82]. Difficulties included navigation issues, software misunderstanding users’ inputs [73,85], or, in other cases, speech recognition problems [81,97]. Some studies also reported software glitches leading to CBS crashing or connection delays [28,75,76]. Despite causing user dissatisfaction, many studies still reported improvements in students’ knowledge, skills, and enjoyment [75,76,85]. Moreover, some students may find the avatar’s appearance and interactions to be unrealistic [28,76,81,82,97]. This phenomenon could be associated with the “uncanny valley” concept, which suggests that as the representation of a human becomes more realistic, the observer’s focus on its flaws paradoxically increases [108].

CBS, similar to other software, must be updated regularly. The obsolescence and longevity issues related to an outdated tool have also been identified in different disciplines [5,109]. This has led to concerns about the need for a new cycle of selection, implementation, and stakeholders’ cooperation, which may deter educators from embracing change [110,111]. However, it is likely these problems are overemphasized in the literature, which primarily discusses novel, cutting-edge products. As commercial versions become available and mature into stable products, these issues are likely to be significantly reduced.

## 5. Future Directions and Suggestions

CBS has the potential to play a more important role in pharmacy practice training. It provides safe, relevant, valuable, and realistic training that students need before joining the workforce—a training that meets the curriculum learning objectives. However, the widespread adoption of CBS in pharmacy education depends on overcoming resistance to change, providing more encouragement from accreditation bodies, and better resourcing from educational providers. Reflecting on our findings in this review, we developed some suggestions for strategies that may help increase CBS uptake.

To increase CBS uptake, institutions and decision-makers must comprehend the educational needs for CBS infrastructure, support its implementation, and foster an organisation culture receptive to change. Aligning program needs with appropriate teaching methods and simulation tools is vital [35]. In Appendix A, it is noted that accreditation bodies have been conservative in endorsing the widespread implementation of CBS [2,18], and simulation initiatives in countries such as Australia are still driven mostly by enthusiasts rather than any kind of policy [112].

The integration of technology in pharmacy education should consider students’ and educators’ beliefs, addressing gaps among low performers and challenging high performers, as well as learners’ language, computer skills, and resource availability [113,114,115]. Educators are advised to undergo a process of careful planning, implementation, and evaluation of technologies, including the consideration of social and cultural aspects [22,115]. This aligns with two studies, that advised selecting the best tools for specific learning goals [22,116]. Successful implementation requires time, support, and commitment from educators and institutions, and it is a mistake to imagine that an off-the-shelf solution can be implemented as a turn-key solution [22]. Moreover, establishing supportive communities and social networks for educators can improve success by providing open access to reliable information, experiences, technological developments [115], skill upgrades [117], and technical support [117,118]. Interestingly, there is an emerging role of “educational technologist” in many secondary and higher education institutions in many schools in Ireland, Finland, Norway, Estonia, the Czech Republic, Denmark, and Malta [119]. Such a dedicated position could also be considered in pharmacy schools to support the educators and assist them in developing and maintaining more scenarios without adversely impacting their workload or time.

To overcome the challenge of engaging students, several strategies can be adopted. Providing educators with tools and training for creating interactive content can lead to more widespread use of simulations. Institutions should offer relevant training to help educators use the technology effectively and develop troubleshooting skills [117]. In addition, communicating the benefits of CBS to students would help them be more engaged in the learning process. Involving students in the design phase can also improve the tool and give them a sense of ownership [28].

CBS has grown from being an experimental training method to an evidence-based, effective learning technology that has proven its potential in other healthcare disciplines and during COVID-19. The use of CBS in pharmacy education enables students to access a broad spectrum of simulated training scenarios in a safe environment, offering them limitless learning opportunities that can be accessed on-demand. This presents an opportunity for pharmacy schools to engage the current and future generations of students and manage their resources effectively while providing high-quality training. Further research is required to explore efficient implementation strategies to best utilize and foster the benefits of this training method.

## 6. Strengths and Limitations

The paper clearly defines its research question, which is to identify potential barriers to the implementation of CBS in pharmacy education. The study utilized a comprehensive literature search of five major databases and used the AACODS checklist for grey literature assessment. Moreover, the use of thematic analysis allowed for a detailed exploration of potential barriers to the integration of CBS in pharmacy practice education. The review identified several potential barriers, including resistance to change, cost, time, usability of software, meeting accreditation standards, motivating and engaging students, faculty experience, and curriculum constraints. The review also provided suggestions for overcoming potential barriers to the integration of CBS in pharmacy practice education, such as addressing academic, process, and cultural barriers and engaging in careful planning, collaboration, and investment in resources and training.

Despite this, the review has some limitations. The studies included in the review did not specifically focus on barriers to implementation. As a result, the extracted barriers are based on indirect evidence or the authors’ perspectives, which may not comprehensively represent all possible challenges faced in implementing CBS in pharmacy education. In addition, the studies included in the review may not be equally representative of different regions, institutions, and cultural contexts, which could lead to an incomplete understanding of the barriers faced in various settings.

Lastly, the review suggests that more research is needed to provide evidence-based support and tested strategies for the implementation of CBS in pharmacy practice education.

## 7. Conclusions

This review highlights the barriers that may impede the successful implementation of CBS in pharmacy practice education. To overcome these obstacles, it is crucial to con- duct further research to identify the most effective implementation strategies, including the involvement of different key stakeholders and exploring the impact of these barriers in various contexts and regions. In addition, improved reporting and sharing of best practices are essential to facilitate the successful implementation of CBS. We expect that by addressing these barriers, we can realize the full potential of CBS in pharmacy practice education and help to ensure that future pharmacists are equipped with the necessary skills and knowledge to provide optimal service.

## Figures and Tables

**Figure 1 pharmacy-11-00086-f001:**
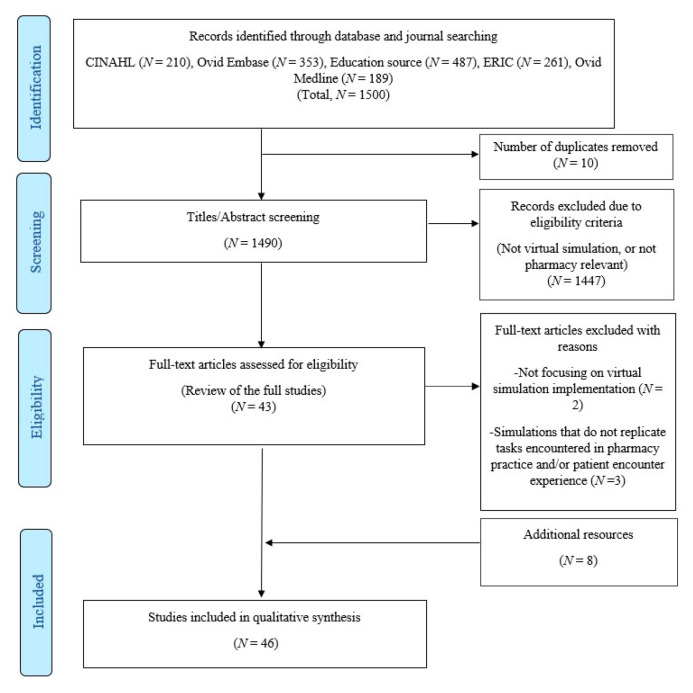
Flow diagram of the study selection process.

**Table 1 pharmacy-11-00086-t001:** Summary of key themes and subthemes processed from the included studies.

Themes	Subthemes per Stakeholder Group		
	Educational Organization	Educators	Students
**Cultural** **barriers**	**Resistance to change** Some organizations are slow in adopting change	**Resistance to change** Considering the new role is too tech-savvy and demanding Skepticism toward the success of this training method	**Resistance to change-**Students’ attitudes
**Process** **barriers**	**Cost** that is often associated with: Lack of needed resources High development costs (or purchase fees) Establishing technical support infrastructure	**Time** For grading and case scenario development, editing or updating training scenarios **Usability of the software** Ease of access and availability of continuous updates for the software	**Time** Needed by students to complete the exercise content **Usability of the software** Software errors/glitches and difficulties in system navigation, data entry and unsuitable interface
**Academic** **barriers**	**Meeting accreditation standards**	**Lack of faculty expertise** **Curriculum constraints**	**Motivating and engaging students** Includes meeting students’ needs and expectations.

## Data Availability

Not applicable.

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
