# Peer review of "Potential Barriers to the Implementation of Computer-Based Simulation in Pharmacy Education: A Systematic Review"

_pharmacy, 2023, doi:10.3390/pharmacy11030086_

Round 1

Reviewer 1 Report

OVERALL:

Thank you for the opportunity to review this manuscript. Overall, this is an interesting topic, and I hope the comments below are helpful in making the manuscript clearer, more concise, and according to PRISMA reporting guidelines.

TITLE:

Narrative is a terminology that can mean many things in the context of reviews. I recognize in the methods what the authors later mean by this, but I would still remove narrative from the title. I’m also questioning whether the ‘potential’ needs to be there, and it could simply read ‘barriers.’

ABSTRACT:

The process used to identify the 42 articles is not specified in the abstract, when it should be, as part of the methods. Overall, the abstract would benefit from a bit of rebalancing, inclusive of less ‘discussion’ and more ‘methods,’ even in the unstructured format. Additionally, the authors should strive to use MeSH terms as keywords, adapting the current ones as appropriate.

INTRODUCTION:

The second paragraph mentioned ‘the apparent potential’ of CBS, yet the paragraph before this does not really provide this, at least not from an outcomes basis. Is there any evidence that educational outcomes are better than the other methods currently in use? This is needed to I think set up the assumption that pharmacy should want to adopt CBS.

I’m not sure that comparing downloads is a sufficient metric, given that the comparable sizes of the medical and pharmacy professionals are quite different. I would also change language from ‘lagging behind,’ which implies a problem, to ‘not yet adopted’ or something similar but more neutral in judgment.

It’s not clear from the introduction what CBS would be specifically useful for in pharmacy. In medicine, use is particularly common for things like anatomy, but this is somewhat less relevant in pharmacy. Perhaps the authors can provide some examples (alongside the efficacy of CBS) that suggest where CBS could be used in pharmacy. I think this is necessary to set the need for this manuscript, which seems to assume we all should be doing this already, without really explaining why.

METHODS:

I’m struggling with the term narrative systematic review. Why not just call it a systematic review? See articles like (https://www.ncbi.nlm.nih.gov/pmc/articles/PMC5137994/ and https://learning.edanz.com/systematic-scoping-narrative/) for an example on why the terminology can be confusing.

The description of the search strategy is insufficient. The authors should include the actual searches as a part of a supplementary index. This will allow reviewers/readers to assess whether appropriate terms were comprehensively searched.

Pharmacy Education (from FIP) is another educationally relevant journal that should be included in such a search.

The methods mention both August 2021 and August 2022 as stop dates for the searches. The authors should clearly identify the date on which the final search was executed.

The inclusion/exclusion criteria of articles within the review is not clearly stated. As a reviewer, I’m unable to understand which criteria were applied. There is a short description that requires much more detail – did analyses have to be in pharmacy schools? Contain certain outcomes? What years of students? What subject areas? What types of study methodologies/papers? Etc.?

The authors should incorporate the bulleted lists in the methods into paragraph format. It’s not usual to have bulleted lists in manuscripts and is disruptive of the section’s flow. This comment applies across the entire manuscript – do not use lists.

More information is needed for how the thematic analysis was performed, as it is essential to the results. How was this operationalized across the team members and what was the iterative process?

RESULTS:

Please fix the “Error! Reference source not found.” in different sections of the results.

References should be attached to all results – for instance, if you say there are 19 studies from the USA, there should be the specific references for those studies within that sentence.

“These simulators were re-ported upon at different stages of their development journey, offering different features and design concepts/methods. As a result of these variations, with the concern that some barriers for one are not necessarily the same as for the other, only the overarching barriers were considered in this exploratory review.” – This information under section 3.1 reads more as methods and should be incorporated in that section.

Under 3.2.1: “It was found to be the most reported barrier in the educators group 51%, followed by 28% for the educational organisation group, and the students group by 21%.” – It’s not clear the source of these percentages. Were these calculated by the authors, from a specific paper? The methods do not describe a quantitative analysis alongside the thematic analysis. This needs better description in the methods for this and following sections.

Overall, the discussions in 3.2.1, 3.2.2 and 3.2.3 are sometimes written in language that suggests the barriers are theoretical, such as “pharmacy educators may not have the technical expertise.” This language should be changed to be clear that specific studies identified this, if this is the case.

DISCUSSION:

“Although there is a strong theoretical basis for the positive impact and benefits of implementing CBS in pharmacy curriculum.” – as mentioned before, the authors have not really demonstrated this, or even provided references to support this.

The subsections of the discussion feel a bit choppy. The results are subsectioned and this makes sense. However, a cohesive discussion section will flow better, with perhaps only subsections for limitations/strengths and future directions, at the most. Things like cost, resistance, and motivation shouldn’t be discussed separately, as they are all integrated and related to each other.

The discussion also requires significant cutting and efforts to be made more concise (probably should be reduced about 50%). Right now, it reads as an extension of the results. It should be framed more in context to existing literature, rather than reciting the studies from the SR itself.

TABLES:

Table 1 is a good idea overall but admittedly it is a bit difficult to interpret on its own. Why do some cells have bullet points and others don’t?

S1 seems like it should be a table in the main manuscript, and not just as supplemental information. It contains the main details of the study and is required for the manuscript to be properly interpreted. There should not be a separate reference list for S1. Please integrate with the rest of the manuscript and have a single reference list.

FIGURES:

The authors are using an older format of PRISMA chart within Figure 1 and should revise according to the most recent published standards.

I don’t believe Figure 2 is needed as the information is not entirely important. If felt necessary to include, a small couple lines of text is sufficient to communicate the trend while removing the actual figure.

There is no need for pie charts in scientific literature – please remove Figure 3. Figures/charts are very useful for showing comparisons or trends, but they are not needed to represent percentages, which are sufficient to be interpreted alone in the text.

Figure 4 can also be deleted as it conveys a limited amount of information. Why is it important to quantify the process barriers? The authors use a thematic analysis, so there seems to be no need for quantitative analysis alongside this.

OTHER:

The authors should include the PRISMA checklist with their submission (for internal review).

Author Response

Reviewer #1:

Overall comment:

Thank you for the opportunity to review this manuscript. Overall, this is an interesting topic, and I hope the comments below are helpful in making the manuscript clearer, more concise, and according to PRISMA reporting guidelines.

Response:

My team and I would like to thank you for your time, efforts, and valuable suggestions. We also appreciate your attention to detail and your dedication to maintaining high-quality standards in research. Please do not hesitate to let us know if you have any further suggestions or concerns.

In the sections below, you will find a detailed point-by-point response to all the comments provided to our team.

Title comment:

Narrative is a terminology that can mean many things in the context of reviews. I recognize in the methods what the authors later mean by this, but I would still remove narrative from the title. I’m also questioning whether the ‘potential’ needs to be there, and it could simply read ‘barriers.’

Response:

Thank you for the feedback to improve our title, we appreciate your comment and your point of view regarding the use of the terms 'narrative, and potential' in our title.

My team and I followed your suggestion of removing the word “narrative” from the title and we tried to explain to the audience what we mean by the term ‘Systematic narrative review’ within the main body.

This systematic narrative approach allowed us to include a broader range of study designs and use narrative synthesis techniques to contextualize our findings, which was essential given the heterogeneity of the included studies, serving our research question and the study aims (Turnbull et al., 2023; Lam et al., 2023; Bellhouse et al., 2021).

We appreciate that systematic narrative reviews are less common, and thus more liable to confuse the audience. But note that they are an accepted approach, and do have their place, see:

  • Turnbull, D., Chugh, R., & Luck, J. (2023). Systematic-narrative hybrid literature review: A strategy for integrating a concise methodology into a manuscript. Social Sciences & Humanities Open7(1), 100381.
  • Lam, G. Y. H., Wong, H. T., & Zhang, M. (2023). A Systematic Narrative Review of Implementation, Processes, and Outcomes of Human Library. International Journal of Environmental Research and Public Health, 20(3), 2485. MDPI AG. Retrieved from http://dx.doi.org/10.3390/ijerph20032485.
  • Bellhouse, C., Newman, L., Bilardi, J.E., Temple-Smith, M. (2021). A systematic narrative review of psychological interventions available in the antenatal period to prepare parents for parenting. Current Psychology. https://doi.org/10.1007/s12144-021-02138-z
  • Altman, D. G. (1986). Summing up. The science of reviewing research, Richard J. Light and David B. Pillemer, Harvard University Press, Statistics in Medicine, 5(3), 289-289. https://doi.org/https://doi.org/10.1002/sim.4780050310
  • Cook, D. A., & Triola, M. M. (2009). Virtual patients: a critical literature review and proposed next steps. Med Educ, 43(4), 303-311. https://doi.org/10.1111/j.1365-2923.2008.03286.x
  • Popay, J., Roberts, H., Sowden, A., Petticrew, M., Arai, L., Rodgers, M., & Britten, N. (2006). Guidance on the conduct of narrative synthesis in systematic Reviews. A Product from the ESRC Methods Programme. Version 1.

Regarding your suggestion to remove the word "potential":

We appreciate your point of view. However, we believe that using the term "potential" is essential so that we do not overstate our findings. Our study identifies a broad range of barriers that may be encountered in implementing computer-based simulations, interpreting these barriers from commentary and discussion within the included papers. This contrasts with more definitive barriers that have been established by deliberate research aimed specifically at determining barriers – to date no such deliberate research exists, and this gap is one of the ultimate findings of this review.

Abstract comment:

The process used to identify the 42 articles is not specified in the abstract, when it should be, as part of the methods. Overall, the abstract would benefit from a bit of rebalancing, inclusive of less ‘discussion’ and more ‘methods,’ even in the unstructured format. Additionally, the authors should strive to use MeSH terms as keywords, adapting the current ones as appropriate.

Response:

Thank you, we have revised and edited the Abstract section accordingly.

Introduction comments:

- The second paragraph mentioned ‘the apparent potential’ of CBS, yet the paragraph before this does not really provide this, at least not from an outcome’s basis.

Is there any evidence that educational outcomes are better than the other methods currently in use? This is needed to I think set up the assumption that pharmacy should want to adopt CBS.

- It’s not clear from the introduction what CBS would be specifically useful for in pharmacy. In medicine, use is particularly common for things like anatomy, but this is somewhat less relevant in pharmacy. Perhaps the authors can provide some examples (alongside the efficacy of CBS) that suggest where CBS could be used in pharmacy.

I think this is necessary to set the need for this manuscript, which seems to assume we all should be doing this already, without really explaining why.

- I’m not sure that comparing downloads is a sufficient metric, given that the comparable sizes of the medical and pharmacy professionals are quite different.

- I would also change language from ‘lagging behind,’ which implies a problem, to ‘not as widely adopted’ or something similar but more neutral in judgment.

Response:

Thank you for bringing this to our attention. We agree with your comments, and have revised and edited this section accordingly.

METHODS comments:

I’m struggling with the term narrative systematic review. Why not just call it a systematic review? See articles like (https://www.ncbi.nlm.nih.gov/pmc/articles/PMC5137994/ and https://learning.edanz.com/systematic-scoping-narrative/) for an example on why the terminology can be confusing.

The description of the search strategy is insufficient.

The authors should include the actual searches as a part of a supplementary index. This will allow reviewers/readers to assess whether appropriate terms were comprehensively searched.

Pharmacy Education (from FIP) is another educationally relevant journal that should be included in such a search.

The methods mention both August 2021 and August 2022 as stop dates for the searches. The authors should clearly identify the date on which the final search was executed.

The inclusion/exclusion criteria of articles within the review is not clearly stated. As a reviewer, I’m unable to understand which criteria were applied. There is a short description that requires much more detail

– Did analyses have to be in pharmacy schools? Contain certain outcomes? What years of students? What subject areas? What types of study methodologies/papers? Etc.?

The authors should incorporate the bulleted lists in the methods into paragraph format. It’s not usual to have bulleted lists in manuscripts and is disruptive of the section’s flow.

This comment applies across the entire manuscript – do not use lists.

More information is needed for how the thematic analysis was performed, as it is essential to the results.

How was this operationalized across the team members and what was the iterative process?

Response:

Thank you for your comments. As seen in our response earlier to the overall comments, we tried to clarify our definition of a systematic narrative review. We have also attached a few examples of similar pieces of literature that have utilized this review style.

We have revised and updated the search strategy adding more details.

Regarding the use of Pharmacy Education (from FIP): This journal is listed in EBSCO and indexed in the Emerging Sources Citation Index (ESCI - Web of Science), EMBASE and SCOPUS. All of which fall under our search strategy.

We want to clarify that our systematic review search employed five major databases (Ovid Medline, CINAHL, ERIC, Education Source, and Ovid EMBASE), and subsequently, we also manually searched two key pharmacy education-focused journals (American Journal of Pharmaceutical Education (AJPE), and Currents in Pharmacy Teaching and Learning (CPT&L)), This was only done as a precaution, to ensure that we captured any further relevant papers.

However, we can see how reporting this step has caused confusion; therefore, we have decided to remove reference to this from the manuscript, since there were no additional studies identified in that step – which is unsurprising on reflection, since these journals are also indexed.

Regarding the search dates: Sorry for this typo, we have revised and updated the manuscript to ensure no further typos are found.

Regarding the inclusion and exclusion criteria: Thank you for bringing this to our attention, we have updated this section accordingly.

Regarding the format and the use of bullet points: as suggested, my team and I have revised and edited the manuscript accordingly.

Results comments:

Please fix the “Error! Reference source not found.” in different sections of the results.

References should be attached to all results – for instance, if you say there are 19 studies from the USA, there should be the specific references for those studies within that sentence.

“These simulators were re-ported upon at different stages of their development journey, offering different features and design concepts/methods. As a result of these variations, with the concern that some barriers for one are not necessarily the same as for the other, only the overarching barriers were considered in this exploratory review.” – This information under section 3.1 reads more as methods and should be incorporated in that section.

Under 3.2.1: “It was found to be the most reported barrier in the educators group 51%, followed by 28% for the educational organisation group, and the students group by 21%.”

– It’s not clear the source of these percentages. Were these calculated by the authors, from a specific paper? The methods do not describe a quantitative analysis alongside the thematic analysis. This needs better description in the methods for this and the following sections.

Overall, the discussions in 3.2.1, 3.2.2 and 3.2.3 are sometimes written in language that suggests the barriers are theoretical, such as “pharmacy educators may not have the technical expertise.”

This language should be changed to be clear that specific studies identified this, if this is the case.

Response:

Regarding Errors: Apologies., we believe the formatting was corrupted when we converted the original manuscript to MDPI style. We have now repaired the tables and references in the revised manuscript.

Regarding references to all results: Thank you for bringing this to our attention. My team and I have revised and edited the manuscript accordingly.

Regarding the language tune: While it is understandable that a specific and decisive identification of a barrier is preferred, this cannot always be done in this review. As we noted, the included studies had a range of different simulators (24 simulators) with various features, applications, and evaluation methods. These simulators were reported upon at different stages of their development, offering different features and design concepts/methods, and were not specifically researching barriers affecting adoption of the technology in the classroom. As a result of these variations, with the concern that some barriers for one are not necessarily the same as for the other, only the overarching barriers were considered in this exploratory review and that’s why the authors were particular in viewing these overarching barriers with a neutral tone and mentioning that future research direction should fill this gap and use this study as a guide to developing further interview and survey questions involving key stakeholders groups. Indeed, this is research we are conducting presently.

Comments regarding 3.2.1, 3.2.2., 3.2.3: We value this guidance and we have revised and edited the manuscript accordingly.

Discussion comments:

“Although there is a strong theoretical basis for the positive impact and benefits of implementing CBS in pharmacy curriculum.”

– as mentioned before, the authors have not really demonstrated this, or even provided references to support this.

The subsections of the discussion feel a bit choppy. The results are sub sectioned and this makes sense. However, a cohesive discussion section will flow better, with perhaps only subsections for limitations/strengths and future directions, at the most.

Things like cost, resistance, and motivation shouldn’t be discussed separately, as they are all integrated and related to each other.

The discussion also requires significant cutting and efforts to be made more concise (probably should be reduced about 50%). Right now, it reads as an extension of the results. It should be framed more in context to existing literature, rather than reciting the studies from the SR itself.

Response:

Thank you for your feedback regarding the benefits of using CBS. This section has been amended and strengthened in the revised manuscript according to the valued feedback received.

Regarding the subsections, we have revised and edited it further as requested.

Regarding how the discussion reads as an extension of the results and is overlong.  We have significantly reduced this section and restructured it so that it does not so closely follow the structure of the results section. We believe this restructuring and tightening allows the higher-level commentary and discussion to emerge more clearly.

Tables comments:

Table 1 is a good idea overall but admittedly it is a bit difficult to interpret on its own. Why do some cells have bullet points and others don’t?

S1 seems like it should be a table in the main manuscript, and not just as supplemental information. It contains the main details of the study and is required for the manuscript to be properly interpreted. There should not be a separate reference list for S1. Please integrate with the rest of the manuscript and have a single reference list.

Response:

Regarding Table 1, Apologies; we have now repaired the tables.

Regarding – table S1: We agree with you. However, the Journal may not wish include a large table as part of the actual paper (instead of in the supplementary material). We will seek advice from the editor.

Figures comments:

The authors are using an older format of PRISMA chart within Figure 1 and should revise according to the most recent published standards.

I don’t believe Figure 2 is needed as the information is not entirely important. If felt necessary to include, a small couple lines of text is sufficient to communicate the trend while removing the actual figure.

There is no need for pie charts in scientific literature – please remove Figure 3.

Figures/charts are very useful for showing comparisons or trends, but they are not needed to represent percentages, which are sufficient to be interpreted alone in the text.

Figure 4 can also be deleted as it conveys a limited amount of information.

Why is it important to quantify the process barriers? The authors use a thematic analysis, so there seems to be no need for quantitative analysis alongside this.

Response:

Thanks for this advice. We have incorporated these suggestions.

Other comments:

The authors should include the PRISMA checklist with their submission (for internal review).

Response:

Good point. We have prepared a PRISMA checklist highlighting our compliance with the required items. We have included the completed PRISMA checklist as a supplementary file.

Reviewer 2 Report

Thank you for the opportunity to review the manuscript Potential barriers to the implementation of computer-based simulation in pharmacy education: A systematic narrative review

The review is well conducted and interesting/needed in the field of pharmacist education.

 I have a few minor comments:

-        - Introduction, second paragraph. I am not sure the comparisons with the medical field (doctors/nurses) is relevant. There are so many more doctors/nurses in the world than pharmacists.

-        -  Three “error-messages”: Second line after heading 3.1, paragraphs before and after heading 3.2.2.

-        -  In the result, please include the real numbers and not just percentages. E.g. start page 2 where it says 60% and, at least for me, the denominator is not clear. The same goes for Figure 3

-        -  Figure 4 – no label on the y-axis.

-        -  Discussion: there are a bit too much repetition of results – please shorten.

-        -  Section 5 – potential barriers – is very helpful IF you want to implement CBS. However, it can be read as if you HAVE TO do this. Please write in a more neutral way. (I know there are advantages, and we use CBS at my university, but it can e.g. be rather expensive and impossible for poorer universities/countries.)

-        -  Write out Generation in Gen Z. At least I (not native English-speaking) had to think for a while before I understood.

Author Response

Reviewer #2:

Comment:

Thank you for the opportunity to review the manuscript Potential barriers to the implementation of computer-based simulation in pharmacy education: A systematic narrative review

The review is well conducted and interesting/needed in the field of pharmacist education.

  • I have a few minor comments:
    • Introduction, second paragraph: I am not sure the comparisons with the medical field (doctors/nurses) is relevant. There are so many more doctors/nurses in the world than pharmacists.
    • Three “error-messages”: Second line after heading 3.1, paragraphs before and after heading 3.2.2.
    • In the result: please include the real numbers and not just percentages. E.g. start page 2 where it says 60% and, at least for me, the denominator is not clear. The same goes for Figure 3
    • Figure 4: no label on the y-axis.
    • Discussion: there are a bit too much repetition of results – please shorten.
    • Section 5 – potential barriers: is very helpful IF you want to implement CBS. However, it can be read as if you HAVE TO do this. Please write in a more neutral way. (I know there are advantages, and we use CBS at my university, but it can e.g. be rather expensive and impossible for poorer universities/countries.)
    • Write out Generation in Gen Z. At least I (not native English-speaking) had to think for a while before I understood.

Response:

Thank you for your thorough review of the manuscript. My team and I appreciate your positive feedback and the constructive comments you provided. Please find below my response to each of your comments:

  • Introduction, second paragraph: Thank you for bringing this up. I agree that the comparison with the medical field may not be entirely relevant in this context. I will revise the paragraph to avoid any confusion.
  • Three "error-messages": Thank you for pointing out these errors. I have carefully reviewed the manuscript and corrected any error messages.
  • Results: I appreciate your suggestion to include the real numbers rather than just percentages. I have updated the manuscript accordingly.
  • Figure 4: Thank you for bringing this to my attention. After careful revision with my team and also considering reviewer one’s comment about this figure, we have decided to delete this figure.
  • Discussion: I appreciate your comment regarding the repetition of results. We have significantly reviewed the discussion section and streamlined the presentation of the results to avoid any unnecessary repetition.
  • Section 5 – Potential barriers: Thank you for your feedback on this section. My team and I agree that it is essential to present the potential barriers to implementing CBS in a neutral manner. We have revised the language to ensure that the information is presented objectively.
  • Write out Generation in Gen Z: Thank you for bringing this to my attention. We have revised and edited the manuscript accordingly to improve the clarity of what we meant by new generation of pharmacy students.

Thank you again for your thoughtful review, and please let me know if you have any additional comments or suggestions.
